# Inside the Black Box: A Narrative Review on Comprehensive Geriatric Assessment-Driven Interventions in Older Adults with Cancer

**DOI:** 10.3390/cancers14071642

**Published:** 2022-03-24

**Authors:** Vincent Thibaud, Claire Billy, Joaquim Prud’homm, Jeanne Garin, Benoit Hue, Catherine Cattenoz, Dominique Somme, Aline Corvol

**Affiliations:** 1Department of Geriatrics, CHU Rennes, Université Rennes 1, 35000 Rennes, France; claire.billy@chu-rennes.fr (C.B.); jeanne.garin@chu-rennes.fr (J.G.); catherine.cattenoz@chu-rennes.fr (C.C.); 2Department of Hematology, CHU Rennes, Université Rennes 1, 35000 Rennes, France; 3Department of Hematology, Hôpital Saint-Vincent, Université Catholique de Lille, 59000 Lille, France; 4Laboratoire Traitement du Signal et de l’Image (INSERM LTSI—UMR 1099), CHU Rennes, Université Rennes 1, 35000 Rennes, France; joaquim.prud-homm@inserm.fr; 5Department of Pharmacy, CHU Rennes, Université Rennes 1, 35000 Rennes, France; benoit.hue@chu-rennes.fr; 6Centre Eugène Marquis, Department of Medical Oncology, Université Rennes 1, 35000 Rennes, France; 7Laboratoire Arènes(CNRS UMR 6051), CHU Rennes, Université Rennes 1, 35000 Rennes, France; dominique.somme@chu-rennes.fr (D.S.); aline.corvol@chu-rennes.fr (A.C.)

**Keywords:** CGA, comprehensive geriatric assessment, cancer, malignant hemopathies, older patients

## Abstract

**Simple Summary:**

Comprehensive geriatric assessment is defined as a multi-dimensional, multi-disciplinary diagnostic and therapeutic process that is conducted to determine the medical, mental, and functional problems that older people with frailty have so that a coordinated and integrated plan for treatment and follow-up can be developed. Progress has been made in the definition of the best way to detect problems, but the benefits are mostly based on prognosis stratification and on the adaptation of cancer treatment. The present review aims to evaluate the level of evidence regarding geriatric interventions proposed following the detection of a problem in cancer patients. This review highlights the scarcity of published studies on this topic. Comprehensive geriatric assessment-based interventions have not yet demonstrated their specific impact. Multi-domain interventions seem promising, especially when they are based on global assessments. However, standardization seems difficult considering the lack of evidence for each domain.

**Abstract:**

There is a consensus that the use of comprehensive geriatric assessment (CGA) is good clinical practice for older patients with solid tumors or hematological malignancies. To be complete, a CGA must include a geriatric assessment and an intervention plan. According to the SIOG consensus, a CGA should assess several domains: functional status, comorbidity, cognition, mental health status, fatigue, social status and support, nutrition, and the presence of geriatric syndromes. Progress has been made in the definition of the best way to detect problems, but the benefits are mostly based on prognosis stratification and on the adaptation of cancer treatment. The present review aims to evaluate the level of evidence regarding geriatric interventions proposed following the detection of a problem in cancer patients in each domain mentioned in the SIOG consensus. An online search of the PubMed database was performed using predefined search algorithms specific for each domain of the CGA. Eligible articles had to have well-defined interventions targeting specific domains of the CGA. We screened 1864 articles, but only a few trials on single-domain interventions were found, and often, these studies involved small groups of patients. This review highlights the scarcity of published studies on this topic. The specific impacts of CGA-based interventions have not yet been demonstrated. Multi-domain interventions seem promising, especially when they are based on global assessments. However, standardization seems difficult considering the lack of evidence for each domain. New studies are necessary in multiple care contexts, and innovative designs must be used to balance internal and external validity. An accurate description of the intervention and what “usual care” means will improve the external validity of such studies.

## 1. Introduction

In the past century, life expectancy in developed countries has substantially increased. The world population is aging, and it is expected that the number of people over 80 years of age will more than double in Europe before the end of the 21st century, from 26.8 million (5.8%) in 2019 to 60.8 million (14.6%) in 2100 [1]. As the incidence of cancer and malignant hemopathies increases with age (approximately 70% of patients with cancer are aged 65 years and older [2]), recommendations for the optimal management of these diseases in the older population are urgently needed. There is a consensus, based on original studies and meta-analyses, that the use of Comprehensive Geriatrics Assessment (CGA) is good clinical practice for most older patients with solid tumors or hematological malignancies, and its implementation is recommended by all major Clinical and Geriatric Oncology Societies [2]. CGA is defined as a multi-dimensional, multi-disciplinary diagnostic and therapeutic process that is conducted to determine the medical, mental, and functional problems that older people with frailty have so that a coordinated and integrated plan for treatment and follow-up can be developed [3]. Thus, to be complete, a CGA must include a geriatric assessment and an intervention plan. In addition, in 2014, the International Society of Geriatric Oncology (SIOG) consensus recommended that the following domains be assessed in a CGA: functional status, comorbidity, cognition, mental health status, fatigue, social status and support, nutrition, and the presence of geriatric syndromes [4].

The CGA was initially used in oncology to distinguish fit, vulnerable, and frail patients [5]. High-quality studies have been published that support the use of CGA in the identification of geriatric syndromes and the prediction of mortality and chemotherapy toxicity. However, these studies generally limit the CGA process to the assessment component, which is mainly tool-based (e.g., cognitive scores, nutritional screening tools, and comorbidity scores) [6,7], and use the scores to adapt the treatment plan. Using the Delphi approach, experts proposed recommendations for implementing CGA-guided care processes [8], but these recommendations were mainly based on the extrapolation of results obtained in a non-cancer population [2]. Previous meta-analyses and review articles focused on the prognostic and predictive value of the CGA in assessing overall survival and adverse outcome rates, and CGA is currently used to prevent over- and under-treatment [9]. It also informs shared decision-making conversations [10], but no clear impact on mortality rates has been demonstrated [11].

As we did not find any article with compelling evidence that focused on the intervention section of the CGA and its efficacy, we decided to review the literature on the “geriatric intervention plan” included in the CGA. We aimed to review the level of evidence of geriatric interventions that was proposed following the detection of a problem in cancer patients in each domain mentioned in the SIOG guidelines. In our review, we only included CGA-based interventional studies regarding older patients (above 65 years old) receiving treatments for cancer or a malignant hemopathy, regardless of the design or the evaluation of the intervention. As the content of interventions has evolved during the last few decades, we limited the research we reviewed to that published in the last twelve years (2010–2022). Our purpose was to identify, domain by domain, the outcomes of the proposed interventions and to highlight associated knowledge gaps that could be addressed in future studies.

## 2. Materials and Methods

A review of the recent literature was conducted to provide an overview of the current knowledge on the impact of CGA-driven interventions on older patients suffering from solid tumors or malignant hemopathies.

The research questions were as follows:Are there strategic CGA interventions for each domain of the CGA?What is the current level of evidence for these interventions?Are there knowledge gaps on these themes that could be filled in future studies?

Comprehensive literature research algorithms for the PubMed website were created by the authors (geriatricians, onco-geriatricians, and a hematologist) for the domains of CGA determined by Wildiers et al. [4]: functional status, comorbidity, cognition, mental health status, fatigue, social status and support, and nutrition. Geriatric syndromes, such as falls, malnutrition, or delirium, were not included in our research algorithm, as these would have been redundant in the abovementioned domains.

The research algorithms consisted of a standardized algorithm and a specific algorithm for each domain of the CGA.

A standard algorithm was designed to identify articles concerning patients suffering from solid tumors or malignant hemopathies who underwent a CGA. It is available in Appendix A. The combined algorithms for each domain are available in Appendix B.

In our screening, we included articles published since 2010 (1 January 2010 to 1 January 2022). We did not include filters on the type of article to avoid excluding relevant publications, but rather screened all of the results of our algorithms. Articles were eligible for inclusion if they were published in English and were original articles describing comparative studies carried out on patients aged 65 years or older who were receiving treatments for solid tumors or malignant hemopathies. The endpoints of interest were overall survival, the completion of chemotherapy, adverse events, and quality of life. We screened for well-defined interventions targeting each specific domain of the CGA in order to review the current evidenced-based data on the impact of interventions targeting each domain.

Every domain of the CGA was attributed to one of the authors. The designated author for each CGA domain reviewed all of the abstracts yielded by the search in terms of their relevance by screening their titles and abstracts. He selected the papers for which the full text was reviewed. Then, he discussed the inclusion (according to the inclusion criteria) and quality of the studies with V.T. In the case of non-agreement with the presence of inclusion criteria, the articles’ full texts were reviewed by a third author (A.C.) to reach a consensus.

Each author responsible for a domain of the CGA assessed the quality of the included study and extracted the relevant data from the included studies.

The findings of this review are presented using a narrative style. First, the results section describes the outcome of the literature search for each domain of the CGA. Then, the characteristics of the selected intervention programs are described to provide insight into the impact of those programs.

## 3. Results

### 3.1. Functional Status

We screened 588 articles related to this domain with our PubMed algorithm and found 2 articles that evaluated the direct effect of non-pharmaceutical treatments administered after a CGA on the functional status of older adults with cancer. No pharmaceutical treatment was identified. Arrieta et al. conducted a prospective, multi-centered, randomized study in people with cancer or hematologic malignancies aged 70 years and older [12]. The physical activity intervention group received one year of training delivered by a professional instructor twice a week, with exercises targeting strength, balance, proprioception, flexibility, and aerobic training. During this training period, the instructor called the participants twice a month during the first 6 months, and then monthly between the 6th and the 12th month to adapt the program to their needs. Three hundred and two adults (mean age: 76), were included in this study. Two years after the onset of the study, there was no difference in the Short Physical Performance Battery, gait speed, International Physical Activity Questionnaire, and verbal fluency scores between groups. The overall survival and hospitalization rates did not differ significantly between the two groups at 1 year (10% vs. 11% and 4% vs. 9%, respectively) and 2 years (20% vs. 20% and 29% vs. 25%, respectively). A posteriori subgroup analysis in female participants showed a positive tendency, with less frequent decline in Short Physical Performance Battery scores in the intervention group (*n* = 37) compared to the usual care group (*n* = 44); (6.2% vs. 21.7%, respectively, *p* = 0.019) and for normal nutritional status at the end of the follow-up in the intervention group (*n* = 31) compared to the usual care group (*n* = 24); (11.1% vs. 24.5%, respectively, *p* = 0.009). No data regarding quality of life were included in this study.

Pergolotti et al. (2019) conducted a randomized controlled trial with 63 adults aged 65 years and older (mean age: 74) with a recent diagnosis or recurrence of cancer within 5 years and at least one functional limitation, as identified with a geriatric assessment [13]. The intervention group received individualized occupational therapy and physical therapy adapted to their needs. Follow-up phone calls were made to ensure occupational/physical therapy appointments were held and to collect final post-assessment data 3 months later. Perceived possibilities for activity, rated with the Possibilities for Activity Scale, were significantly improved in the intervention group (evaluated with the Possibilities for Activity Scale scores (Δ = 3.11 in the intervention group and Δ = −3.2 in the control group, *p* = 0.04). However, the functional status, as evaluated with the Nottingham Extended Daily Living Score and the Patient-Reported Outcomes Measure Information System, significantly declined in both the intervention and control groups, but this decline did not differ between groups.

### 3.2. Comorbidity and Polypharmacy

We screened 834 articles related to this domain with our PubMed algorithm. Among these articles, only one study evaluated the impact of comorbidities and polypharmacy-based interventions [14]. Regarding polypharmacy assessment, three assessment tools are commonly used to evaluate potential inappropriate medications or potential adverse events: the Beers criteria [15], the START/STOPP tool [16], and the Medication Appropriateness index [17]. The conducted interventions regarding comorbidity and polypharmacy domains include referral to other specialists, complementary exams, and a review of prescriptions at the pharmacy to optimize dosages for patients and discontinue inappropriate medications.

Whitman et al. evaluated the feasibility of a pharmacist-led medication and deprescribing intervention. The authors used a combination of the three tools described above to identify inappropriate medications. In this pilot study of 26 patients (mean age: 81) suffering from various types of cancer [14], the well-described intervention led to the deprescription of 73% of potentially inappropriate medications, resulting in a mean of three (0–12) medications deprescribed per patient. Based on the University Health System Consortium outcomes cost data, the results of this intervention could lead to a reduction in healthcare expenditures by more than USD 4000 per patient. Patient-related outcomes also included a reduction in symptoms after the intervention for sixteen patients (60%), which may have improved quality of life and observance within this group of patients. The authors did not report any effect on mortality or chemotherapy completion.

### 3.3. Cognition and Mental Health Status

Among the 107 articles screened with our PubMed algorithm related to this domain, we found no intervention that specifically targeted cognition or mental health in older cancer patients. Numerous interventions offered by nurses, psychologists, or peers that targeted depression or anxiety in cancer patients were evaluated. They recommended psychoeducation, psychotherapy, or physical activities (walking, mind–body practice, etc.). These studies were not included in the present analysis, as no CGA was performed, and interventions were not specifically designed for patients over 65 years old [18,19]. Cognition has a high impact on the feasibility of a care plan, and its assessment is crucial in the decision-making process. Nonetheless, we did not find an intervention that targeted this status in patients with active cancer. When these syndromes are identified with a CGA, numerous adaptations can be recommended: social support, psychological support, occupational therapy, medication review, home care plan adaptation, referral to a psychiatrist, psychotropic drug therapy, etc. However, some observational studies show that there may be a gap between assessment and intervention. In the study by Trevino et al. [20], psychological and social support was only offered to 25% of patients who were identified to have elevated distress or poor social support. Malik et al. revealed that cognitive screening allowed for the identification of increased delirium risks for surgical patients, but the impact of preventive strategies was not evaluated in this study [21].

### 3.4. Fatigue

Fatigue is a subjective experience, and several medical disorders may contribute to it. Cancer-related fatigue may be qualitatively different from tiredness; usually, it is not relieved by rest. We screened 31 articles related to this domain with our PubMed algorithm, but no article matched our inclusion criteria that targeted the impact of fatigue-oriented interventions in older patients suffering from malignant pathologies. The few existing studies that evaluated fatigue interventions combined them with a more global intervention package based on the CGA (see the Global Intervention section below). As fatigue may affect behavioral, cognitive, somatic, and affective domains, its assessment and management overlap with other CGA domain assessments.

### 3.5. Social Status and Support

We screened 113 articles related to this domain with our PubMed algorithm but found no intervention that specifically targeted social status and support for the improvement of cancer prognosis. The French PREDOMOS study aimed to evaluate an intervention that included social aids and techniques using domotic and remote assistance for the improvement of quality of life of older individuals treated for cancer who were isolated or at risk of isolation [22]. Unfortunately, it was stopped because of recruitment difficulties.

### 3.6. Nutrition

We screened 222 articles related to this domain with our PubMed algorithm. Among these articles, only one study assessed the impact of nutritional interventions [23]. Regarding nutritional assessment, the most commonly used tools are the Mini Nutritional Assessment, Body Mass Index, and weight loss. Nutritional interventions included the use of nutritional supplements, referrals to a dietitian, diet recommendations, oral care, physical/occupational therapy for food intake problems, plans for dentures, and appetite stimulants. Only one article evaluated the impact of interventions targeting nutrition [23]. Bourdel-Marchasson et al. evaluated the impact on mortality of a nutritional intervention consisting of diet counseling with the aim of achieving an energy intake of 30 kCal/kg body weight/day and 1.2 g protein/kg/day in a randomized controlled trial that involved 336 older patients receiving chemotherapy for solid or hematological malignancies. This study showed a significant increase in the dietary intake for the intervention group (*p* < 0.01) but no difference in the 1-year (*p* = 0.74) or 2-year (*p* = 0.37) survival or in grade 3–4 infections. One-year and two-year mortality rates were similar in both groups (R = 1.1, 95% CI = 0.8–1.5, *p* = 0.74, and RR = 1.1, 95% CI = 0.9–1.5, *p* = 0.37, respectively). The response to chemotherapy was also similar between groups. No data on quality of life were included in this study.

### 3.7. Subsection Global Interventions

During our review of CGA-based interventions for each domain, we found seven articles that evaluated interventions based on a more global CGA: four observational studies and three randomized controlled trials. These articles did not evaluate a specific intervention, nor interventions relating to a specific domain of the CGA. Instead, they assessed the impact of global interventions based on the CGA. There were multiple interventions for every domain, and several domains were usually targeted. The implementation of the intervention protocol differed from one study to the other. The CGA evaluator determined interventions beforehand. Interventions were implemented either by a geriatrician/CGA multi-disciplinary team or by an oncologist, with or without a follow-up.

The observational study (*n* = 135) published by Kalsi et al. on the effect of a geriatric assessment and intervention plan compared to usual care showed an improvement in the chemotherapy completion rate (33.8% vs. 11.4%, *p* = 0.006) but no significant effect on grade 3+ toxicity rate (43.8% vs. 52.9%, *p* = 0.292) [24]. In a follow-up study of 714 patients suffering from various solid cancers, Ørum et al. showed a potential reduction in the mortality rate in the CGA-driven intervention group, but the study failed to reach significance (*p* = 0.05) [25]. Concerning hematological malignancies, Derman et al. published an observational study focused on older patients receiving stem cell transplantation. The survival rates were evaluated before (2005–2012) and after (2013–2018) the implementation of a CGA-guided multi-disciplinary team clinic. The survival rate appeared higher in the latter group, but these results may be considered as biased due to the presence of a historical comparative group and careful patient selection over the years [26]. These studies did not present data related to quality of life.

Three recently published, randomized controlled trials, namely, the GAIN, GERICO, and GAP 70+ studies [27,28,29], evaluated the impact of CGA-driven interventions on chemotherapy toxicity. In these three studies, comorbidities and polypharmacy interventions were not separated from other CGA domain interventions. In the GAIN study, Li et al. demonstrated that a non-standardized, CGA-based intervention, delivered by a multi-disciplinary team, led to a significant reduction of more than 10% in terms of the toxic effects related to grade 3 or higher chemotherapy compared to the standard of care arm (*n* = 613, 50.5% vs. 60.6%, *p* = 0.02). This result was achieved with similar treatment intensity in the two groups. In the GAP70+ study, Mohile et al. obtained similar results regarding toxicity in a cluster-randomized trial (*n* = 718, 51% vs. 71%, *p* = 0.001). In the trial, geriatric intervention was lighter (only recommendations addressed to the oncologist) but included the modification of the chemotherapy treatment plan. Therefore, it is unclear whether toxicity reduction is a result of superior management or only due to the decreased intensity of treatment. In the GERICO trial, Lund et al. focused on colorectal cancer in a smaller sample of patients. The trial showed that geriatric intervention delivered by a geriatrician improved chemotherapy completion (45% vs. 28%, *p* = 0.0366). An effect on quality of life was also described in a posteriori analysis. The results of these three randomized trials are summarized in Table 1.

## 4. Discussion

This narrative review presented evidence concerning the management section of the CGA, a geriatric assessment tool for older patients with cancer, and highlighted the scarcity of published studies on this topic. A few trials concerned single-domain interventions, and often, these studies involved small groups of patients. Two randomized studies, which tested physical exercise and physical or occupational therapy aiming to improve functional status, respectively, did not lead to statistically or clinically significant results [12,13]. A pilot study demonstrated the feasibility of a pharmacist-led deprescribing intervention with a clear decrease in potentially inappropriate prescriptions, but the study did not evaluate clinical endpoints [14]. In one randomized trial [23], diet counseling led to an increased intake with no effect on mortality rates. No specific interventions that targeted cognitive deficits, fatigue, or social support were identified. The descriptions of the interventions that were carried out were often incomplete, as some authors focused on professionals involved in the study [20], while others focused on precise protocols for domains such as physical exercise or medication reviews [13,14].

The implementation of a CGA in clinical practice and the number of publications on this subject has greatly increased during the last few decades. As “usual care” evolves with new evidence, we limited our research to research published since 2010 to limit heterogeneity in control groups. We chose to develop research algorithms for our narrative review, even though algorithms are not commonly used in narrative reviews, to ensure the better objectivity of our work. In our research algorithm, we required the presence of the term “geriatric assessment”. This means that we did not consider mono-domain interventions in the absence of an initial global geriatric assessment in order to avoid confusing results. This choice may have led us to reject potentially relevant articles, such as the study by Gilbert et al., which showed that perioperative nutritional management overseen by a geriatrician and a dietician (without a CGA) improves adherence to international guidelines for older patients scheduled for colorectal cancer surgery, with no significant impact on adverse events [30]. A study by Ommundsen et al., which was a monocentric trial regarding preoperative geriatric assessment and global intervention, was also not selected, as no specific geriatric domains were cited in the title and abstract [31]. In this study, geriatric intervention failed to reduce grade II–V complications in frail, older patients scheduled for colorectal cancer surgery. This may be linked with the absence of teamwork, with a unique “medical doctor specializing in geriatric medicine” who carried out one session evaluation and intervention based on comorbidity management, a drugs review, and recommendations to prevent delirium and malnutrition. We focused on patients over 65 years old, even though 60-year-old patients are sometimes considered as “older patients” in the oncology literature [32]. As shown here, the specific impacts of CGA-based interventions on cancer patients have not yet been demonstrated, although some studies have shown positive results for older people in geriatric care settings. Therefore, the use of the CGA in a different population may be questionable. Symptoms such as fatigue, weight loss, or depression may be associated with various physiopathological processes and prognoses, and thus require different interventions according to age, comorbidities, and frailty. Some domains assessed in the CGA are more commonly used to select patients and adapt treatment than to develop interventions. For instance, the efficacity of interventions dedicated to cognitive impairment or fatigue are more difficult to achieve than those concerning other CGA domains, as shown in our results. Nevertheless, the efficacities of intervention protocols such as the HELP program in the prevention of delirium in hospitalized older patients have been demonstrated [33]. Therefore, interventional studies remain necessary so that their impact on older patients with cancer and on a treatment’s cognitive toxicity or chemo brain can be evaluated. Other interventions could be designed to reduce adverse events and optimize the adherence of patients with cognitive impairment (e.g., optimizing home intervention professionals, organizing relief periods to prevent helpers’ exhaustion, daily or multi-daily home nurses’ interventions, home calls, and electronic, customized devices to help individuals to remember to take drugs and evaluate the observance (e.g., a communicating pill box)).

Multi-domain interventions seem promising, especially when they are based on global assessments. Three recent, good-quality clinical trials assessed multi-domain interventions on chemotherapy toxicity: GAIN [27], GERICO [28], and GAP70+ [29]. GAIN was the most convincing trial, with a CGA-based intervention characterized by predetermined assessment thresholds. A multi-disciplinary team trained in geriatrics, which included a geriatric nurse practitioner in charge of follow-ups, delivered it. This study was strongly focused on the management section of the CGA. In the control group, the oncologist had access to the results of the CGA (as in the intervention arm) but implemented fewer interventions (12.5% of identified interventions were implemented, versus 76.8% in the intervention arm). The present review highlighted the heterogeneity of possible interventions, control groups, and endpoints. Considering the variability of patients included in these studies, a meta-analysis cannot be undertaken.

As shown in Table 1, both the intervention and control groups were quite different in these three trials. The intervention was either delivered by a team (GAIN), a geriatrician alone (GERICO), or an oncologist informed by recommendations derived from the CGA (GAP70+). The control groups were also different, with no CGA in the GERICO trial and a complete CGA sent to the oncologist in the GAIN trial. So, the control group in the GAIN trial seemed similar to the intervention group in the GAP70+ trial. The authors tried to base the interventions on guides [29] or thresholds [27], but standardization seemed difficult to implement considering the lack of evidence for each domain. The ongoing study “PREPARE”, which aims to evaluate, in a randomized controlled trial, the impact of CGA intervention programs in older adults with cancer, addressed this issue with a geriatric care protocol written by an independent committee and a mandatory centralized nurses’ training program to standardize intervention programs [34].

We note that some endpoints are “oncologic” (e.g., treatment completion, toxicity, and recurrence time) and others are “geriatric” (e.g., functional or nutritional status and deprescription). The CGA can be used to tailor interventions, but its components are sometimes valuable endpoints (e.g., functional or cognitive status). We believe that global endpoints, including the impact of interventions on cancer evolution or patient-centered goals, should be preferred. Overall mortality rates, quality of life, or time spent in hospitals seem interesting outcomes. Concerning ongoing studies, the authors of the PREPARE study [34] choose a co-primary endpoint encompassing overall survival rates and health-related quality of life, whereas Røyset et al. preferred physical function, with quality of life as a secondary objective [35].

These points underline the issues that can arise when designing CGA-based intervention trials. Older patients with cancer usually present several altered domains, limiting the clinical impact of monodomain compared to multi-domain interventions. Monodomain interventions do not seem to be efficient enough, unless the population is highly selected, which decreases the external validity. In this context, two ongoing randomized controlled trials opted for a selected population of patients with colorectal cancer to evaluate a monodomain intervention for patients addressed for surgery [32] or a multi-domain intervention for patients addressed for radiotherapy [35].

The management section of a CGA is a complex intervention that cannot be completely standardized, unlike the assessment section. Randomized controlled trials have been designed to evaluate new drugs, but they may not be the best way to evaluate complex interventions that are highly variable in clinical settings, depending on the provider and the context [36]. In our review, we chose to include observational studies that are frequently more important sources of knowledge in gerontology than randomized controlled trials. Randomization, if chosen, should be clustered as much as possible to avoid contamination bias. Researchers may thus have to make compromises between internal and external validity. Internal validity refers to the reliability of the results and requires a strong standardization of interventions, as well as so-called “usual care”. External validity refers to the generalizability of results and implies flexible real-life practices and the accurate description of contexts [37]. Stepped wedge trial designs seem promising to achieve both internal and external validity. This design allows for the progressive implementation of innovation in different settings, together with its evaluation [38], whereas randomization can be ethically questionable, as the comprehensive geriatric assessment is currently largely recommended. This ethical concern could lead to a recruitment bias in randomized trials if clinicians have access to the CGA outside clinical trials and prefer not to randomize the frailest patients. Stepped wedge trials remain difficult to implement and costly, as they require the collaboration of multiple centers of inclusion. However, it is not impossible in the geriatric field, as demonstrated in the study by Gilbert et al. [30].

## 5. Conclusions

The CGA was first introduced in oncology to distinguish between fit, vulnerable, and frail patients. This approach, sought by oncologists to adapt treatment plans, is complementary with geriatricians’ perspectives that aim to care for rather than to select patients. However, if the CGA’s value in the adaptation of treatment plans is well established, our review stresses the lack of evidence concerning CGA-based interventions for cancer patients. To date, no randomized trial has demonstrated the effect of an intervention on major outcomes such as survival rate or quality of life, despite limited evidence in favor of better chemotherapy completion. Relying on evidence obtained in other care contexts seems inadequate, considering the specificity of symptoms directly linked to oncologic pathologies. New studies are necessary in multiple care contexts, and innovative designs must be used to balance internal and external validity. An accurate description of the intervention and what “usual care” means will improve the external validity of such studies.

## Figures and Tables

**Table 1 cancers-14-01642-t001:** Comparison of the three randomized trials on global CGA-based interventions.

References	Population (Age, Type of Cancer)	Number of Patients	Type of Study	CGA Intervention	Results	
					Primary Outcome	Secondary Outcomes
Li et al. [27] GAIN	65+, solid cancer	613	RCT Single-center	Intervention and referrals, based on predetermined thresholds, delivered by a multi-disciplinary team (oncologist, nurse practitioner, social worker, physical/occupation therapist, nutritionist, and pharmacist) Follow-up by the geriatric nurse practitioner Control group: CGA is sent to the oncologist	AE grade 3–5: reduction in CGA group of 10.1% (95% CI 1.5–18.2; *p* = 0.02)	More completion of chemotherapy treatment plan in the CGA group: 28.4% vs. 13.3%, *p* < 0.01
Lund et al. [28] GERICO	70+, colorectal cancer	142	RCT Single-center	Intervention and referrals, tailored by a geriatrician Follow-up after two months or more frequently Control group: no CGA is performed; no change in chemotherapy treatment	Completion of chemotherapy treatment plan: 45% vs. 28%, *p* = 0.00366.	Less AE grade 3–5 in the CGA group: 28% vs. 39%, *p* = 0.156
Mohile et al. [29] GAP70+	70+, incurable solid tumors or lymphoma	718	Cluster-randomized trial	Geriatric assessment summary and management recommendations (including dose reduction) sent to the oncologist Control group: oncologists received alerts for impaired depression or cognitive score	Relative risk of AE grade 3–5 in CGA group of 0.74 (95% CI 0.64–0.86; *p* = 0.0001)	

Abbreviations: RCT: randomized controlled trial, CT: controlled trial, AE: adverse event.

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
