# Peer review of "Inside the Black Box: A Narrative Review on Comprehensive Geriatric Assessment-Driven Interventions in Older Adults with Cancer"

_cancers, 2022, doi:10.3390/cancers14071642_

Round 1
Reviewer 1 Report
The review from Thibaud and colleagues explore a very interesting topic and is highly relevant and needed. It aims at evaluating the level of evidence of the geriatric interventions in cancers’ patients following CGA. It is well written and clear.
Major comments:
- The authors should clearly state that focusing on each geriatric domain separately, even if scientifically relevant, doesn’t seem to be clinically relevant, with patient usually presenting several altered domains and need multidomain interventions. Monodomain intervention doesn’t seem to be possibly efficient unless the population is highly selected which decrease the external validity.
- Some domains are more used to select patient or adapt treatment than to create a corrective intervention. Searching for an efficient intervention in a single domain such as cognitive status seems not relevant. What could be an efficient intervention for cognitive impairment, even in non-cancer patient?
- It’s not clear why the authors decided to select article after 2010.
- Regarding nutrition domain, several articles that looked at nutritional intervention in cancer patient are missing. I do understand that this due to the selection process, where only study with CGA before intervention and in older patients were taken. However, I feel that it rules out studies that could help the message of the review. In general, and in every domain, articles that can help the reader understand if an intervention could be useful should be cited at least in the discussion.
- Some articles seem to be missing (DOI: 3390/nu13072347 ; DOI: 10.1111/codi.13785 )
- The authors stated: “In our research algorithm, we required the presence of the term “geriatric assess ment”. It means that we did not consider mono-domain interventions in the absence of an initial global geriatric assessment in order to avoid confusing results.” If the subject of the review is to understand what kind of intervention is efficient for older cancer patient, why limiting to patient with CGA and not look at mono-domain intervention in absence of CGA?
- It is true that interventions are highly heterogenous and RCT rare in older cancer patient, but it would be interesting to propose some solutions in the review. The proposed “Stepped wedge trial designs” even though interesting, seems complicated in oncogeriatry.
Minor comments:
- References are missing in line 77 regarding the impact on mortality (ref 10 is about conversation, but not mortality)
- The authors stated (line 390): “CGA has first been introduced in oncology to distinguish fit, vulnerable and frail patients. This approach, sought by oncologists to adapt treatment plans, conflicts with geriatricians’ perspectives that aim to care rather than to select”: but does it conflict or is it complementary?
- Ongoing trial should be cited: DOI: 1186/s12885-016-2927-4 ; DOI: 10.1186/s12876-018-0754-6 ; DOI: 10.1016/j.jgo.2021.11.001 .
Author Response
The review from Thibaud and colleagues explore a very interesting topic and is highly relevant and needed. It aims at evaluating the level of evidence of the geriatric interventions in cancers’ patients following CGA. It is well written and clear.
We thank the reviewer for their relevant comments on the manuscript.
We modified and updated the manuscript according to the reviewers’ comments.
We hope the modifications performed in the manuscript will meet your expectations.
Major comments:
1/ The authors should clearly state that focusing on each geriatric domain separately, even if scientifically relevant, doesn’t seem to be clinically relevant, with patient usually presenting several altered domains and need multidomain interventions. Monodomain intervention doesn’t seem to be possibly efficient unless the population is highly selected which decrease the external validity.
à We agree with that comment. Modifications were added line 383
“Older patients with cancer usually present several altered domains limiting the clinical impact of monodomain compared to multidomain interventions. Monodomain interventions does not seem to be efficient enough unless the population is highly selected which decease the external validity. In this context, two ongoing randomized controlled trial opted for a selected population of patients with colorectal cancer to evaluate monodomain intervention for patients addressed for surgery [35] or multidomain for patients addressed for radiotherapy[34]”
2/Some domains are more used to select patient or adapt treatment than to create a corrective intervention. Searching for an efficient intervention in a single domain such as cognitive status seems not relevant. What could be an efficient intervention for cognitive impairment, even in non- cancer patient?
àWe took this comment in account for the discussion. Modifications were added line 338
« Some domains assessed in CGA are used to select patients and adapt treatment, more than to develop interventions. For instance, efficacity of interventions dedicated on cognitive impairment or fatigue are more difficult to achieve than on other CGA domains as showed in our results. Nevertheless, intervention protocols such as the HELP program have demonstrated the efficacity in prevention of delirium in hospitalized older patients[32]. Therefore, interventional studies remain necessary to evaluate their impact on older patients with cancer and also on treatment’s cognitive toxicity or chemobrain. Other interventions could be designed to reduce adverse events and optimize adherence on patients with cognitive impairment (e.g. optimizing home interventions professionals, organizing relief periods to prevent helpers’ exhaustion, daily or multi-daily home nurses’ interventions, home calls, electronic customed devices to remember to take drugs and evaluate the observance (e.g. communicating pill box)).”
3/It’s not clear why the authors decided to select article after 2010.
à We explained that choice in the discussion, line 312
“The implementation of CGA in clinical practice and the number of publications on this subject has greatly evolved during the last decades. As “usual care” evolves with new evidence, we limited our research to publications since 2010 to limit heterogeneity in control groups.”
4/ Regarding nutrition domain, several articles that looked at nutritional intervention in cancer patient are missing. I do understand that this due to the selection process, where only study with CGA before intervention and in older patients were taken. However, I feel that it rules out studies that could help the message of the review. In general, and in every domain, articles that can help the reader understand if an intervention could be useful should be cited at least in the discussion.
Some articles seem to be missing (DOI: 3390/nu13072347 ; DOI: 10.1111/codi.13785 )
We thank the reviewer for these references. We added them in the discussion part and explained why they were not retrieved by our algorithm, line 318.
“This choice may have led us to left aside potentially relevant articles, such as Gilbert et al study which shows that a perioperative nutritional management by a geriatrician and a dietician (without CGA) improve adherence to international guidelines for older patients scheduled for colorectal cancer surgery, with no significative impact on adverse events[30]. Ommundsen et al study, a monocentric trial on preoperative geriatric assessment and global intervention were also not selected, as no specific geriatric domain were cited in the title and abstract[31]. In this study, geriatric intervention failed to reduce grade II-V complications in frail older patients scheduled for colorectal cancer surgery. This may be linked with the absence of team work, with a unique “medical doctor specializing in geriatric medicine” who carry out a one session evaluation and intervention based on comorbidities management, drugs review, and recommendations to prevent delirium and malnutrition.”
5/ The authors stated: “In our research algorithm, we required the presence of the term “geriatric assessment”. It means that we did not consider mono-domain interventions in the absence of an initial global geriatric assessment in order to avoid confusing results.” If the subject of the review is to understand what kind of intervention is efficient for older cancer patient, why limiting to patient with CGA and not look at mono-domain intervention in absence of CGA?
We choose to focus on CGA driven intervention. It seems us that a narrative review on all type of interventions aside specific treatment, whatever the type of cancer and hemopathies, would have far too large.
To make it clear, the title was change for: “Inside the black box. A narrative review on Comprehensive Geriatric Assessment driven interventions in older adults with cancer”
6/ It is true that interventions are highly heterogenous and RCT rare in older cancer patient, but it would be interesting to propose some solutions in the review. The proposed “Stepped wedge trial designs” even though interesting, seems complicated in oncogeriatry.
We agree with that comment and developed our thought in the discussion.
“In our review, we choose to include observational studies that are frequently more important sources of knowledge in gerontology than randomized controlled trials. Randomization, if chosen, should be clustered as much as possible to avoid contamination bias.”(line 395)
“This design allow progressive implementation of innovation in different setting, together with its evaluation [38], when randomization can be ethically questionable as comprehensive geriatric assessment is currently largely recommended. This ethical questioning could lead to a recruitment biased in randomized trials, if clinicians have access to CGA outside clinical trials, and prefer not to randomized the frailest patients. Stepped wedge trial remain difficult to implement and costly, as it requires the collaborations of multiple centres of inclusion. Thus, il is not impossible in the geriatric field as demonstrated by Gilbert et al study[30].”(line 403)
Minor comments:
7/ References are missing in line 77 regarding the impact on mortality (ref 10 is about conversation, but not mortality)
à As suggested, we added a reference
Li, D.; Soto-Perez-de-Celis, E.; Hurria, A. Geriatric Assessment and Tools for Predicting Treatment Toxicity in Older Adults With Cancer. Cancer J 2017, 23, 206–210, doi:10.1097/PPO.0000000000000269.
8/The authors stated (line 390): “CGA has first been introduced in oncology to distinguish fit, vulnerable and frail patients. This approach, sought by oncologists to adapt treatment plans, conflicts with geriatricians’ perspectives that aim to care rather than to select”: but does it conflict or is it complementary?
à A suggested, we changed conflict for complementarity.
9/Ongoing trial should be cited: DOI: 1186/s12885-016-2927-4 ; DOI: 10.1186/s12876-018-0754-6 ; DOI: 10.1016/j.jgo.2021.11.001 .
à We thank the reviewer for this suggestion. Ongoing trials were added in the discussion part lines 379 and 387
“Concerning ongoing studies, the authors of PREPARE study [33] choose a co-primary endpoint encompassing overall survival and Health –related quality of life, when Røyset et al preferred physical function, with quality of life as secondary objectives [34].”
“In this context, two ongoing randomized controlled trial opted for a selected population of patients with colorectal cancer to evaluate monodomain intervention for patients addressed for surgery [35] or multidomain for patients addressed for radiotherapy[34].”
Reviewer 2 Report
Well-written and clear article but needs to either be re-formatted as a systematic review or undergo significant changes in formatting.
The authors clearly need to define whether this was a systematic review, a narriative review or a scoping review. I note the correct inclusion of search strategy and rationale, but this needs to be explicitly stated.
However, at the moment, it is part systematic review, part narriative review. Please address this
If a systematic review as suggested by the use of the search strategy, I would encourage the authors to submit a STROBE statement in the appendix and change the manuscript to clearly reflect this. It is clear that a deal of work went into this search, but parameters such as PICO (Population, Intervention, comparison, Outcome) need to be clearly stated in the methods. The methods do not provide enough detail on exclusion and exclusion criteria; perhaps a chart would help ?
It is written that authors discussed article inclusion - were they screened by 2 reviewers ? Please give more detail on this process
Title: "Comprehensive Geriatrics Assessment: Inside the black box. 2 Assessing the evidence for strategic interventions for older patients with cancer." - Please remove the "s" from Geriatric"s" and remove black box - doesn't adequately reflect CGA - suggest changing title to "Comprehensive Geriatric Assessment in Older Adults with Cancer: A Scoping Review" or "Comprehensive Geriatric Assessment in Older Adults with Cancer: A Narriative Review"
Abstract: again remove "s" from geriatric"s" and cancers treatment - should just be "cancer treatment"
Introuduction: "is good clinical practice" not "is a good clinical practice"
Author Response
Well-written and clear article but needs to either be re-formatted as a systematic review or undergo significant changes in formatting.
We thank the reviewer for their relevant comments on the manuscript.
We modified and updated the manuscript according to the reviewers’ comments.
We hope the modifications performed in the manuscript will meet your expectations.
1/ The authors clearly need to define whether this was a systematic review, a narrative review or a scoping review. I note the correct inclusion of search strategy and rationale, but this needs to be explicitly stated.
However, at the moment, it is part systematic review, part narrative review. Please address this
If a systematic review as suggested by the use of the search strategy, I would encourage the authors to submit a STROBE statement in the appendix and change the manuscript to clearly reflect this. It is clear that a deal of work went into this search, but parameters such as PICO (Population, Intervention, comparison, Outcome) need to be clearly stated in the methods. The methods do not provide enough detail on exclusion and exclusion criteria; perhaps a chart would help ?
à Our review is narrative, so we modified the text in order to make it clear for the reader.
We modified the title of the manuscript to explicitly add narrative review. We eliminate the word “systematic review” of all our paper to be clear that we do not pretend having performed a systematic review. We put the research algorithm in appendix and changed a sentence in the introduction:
“As we did not found any article compelling the evidence focusing on the intervention section of the CGA and its efficacy, we propose to review the literature on the “geriatric intervention plan” included in the CGA”
We also precised in the method part
“The researchs questions were as follows:
- Is there strategic CGA interventionsfor each domain of CGA?
- What is the the current level of evidence of these interventions?
- Is there knowledge gaps on these themes that could be filled in future studies?”
“The findings of this review are presented using a narrative style. The results section, for each domain of CGA, will first describe the output of the literature search. Then, the characteristics of the selected intervention programs are described to provide insight in the impact of those programs”
We choose to write a narrative review as not enough resources were available to perform a systematic or a scoping review. Our narrative review does not meet the PRISMA criteria for a systematic or a scoping review. A review following those standards may improve and add references to our work but we remain confident in our results. We read and reviewed all discussions of the referenced articles without finding relevant data questioning our conclusions.
We chose to develop research algorithms for our narrative review even though not commonly used in narrative algorithms to ensure a better objectivity of our work.
Before the beginning of our work, we contacted the editor to ensure that our proposed article of narrative review fall in the scope of the special issue of Cancers. As directly mentioned in my email to the assistant editor Mr Xu, in the first paragraph of the Discussion, the word “narrative” was replaced by “systematic” by our translator while reviewing the article and we missed to correct it in the first version of the manuscript.
Exclusion and inclusion criteria are detailed lines 116-125. We also precises line 132.
“Each author responsible for a domain of CGA assessed the quality of the included study and extracted the relevant data from the included studies.”
2/ It is written that authors discussed article inclusion - were they screened by 2 reviewers ? Please give more detail on this process
à As suggested, we gave more details on this process;
“Every domain of CGA was attribuated to one of the authors. The designed author for each CGA domain reviewed all abstracts yielded by the search for their relevance by screening their titles and abstracts. He selected the papers for which the full text was reviewed. Then he discussed the inclusion (according to the inclusion criteria) and quality of the studies with V.T.. In case of non-agreement with the presence of inclusion criteria, the articles full texts were reviewed by a third author (A.C.) to reach consensus.”
3/ Title: "Comprehensive Geriatrics Assessment: Inside the black box. 2 Assessing the evidence for strategic interventions for older patients with cancer." - Please remove the "s" from Geriatric"s" and remove black box - doesn't adequately reflect CGA - suggest changing title to "Comprehensive Geriatric Assessment in Older Adults with Cancer: A Scoping Review" or "Comprehensive Geriatric Assessment in Older Adults with Cancer: A Narrative Review"
We would like to keep the idea of “black box” that resume the fact that CGA and geriatric plans can recover very different realities. We propose
“Inside the black box. A narrative review on Comprehensive Geriatric Assessment driven interventions in older adults with cancer”
4/ Abstract: again remove “s” from geriatric”s” and cancers treatment – should just be “cancer treatment”
à Thank you for that correction
5/ Introduction: "is good clinical practice" not "is a good clinical practice"
à Thank you for that correction
Round 2
Reviewer 1 Report
All comments have been addressed.
Just some minor comments on English language that can be corrected by the authors.
Minor comments:
Line 383 :
Monodomain interventions do not seem to be efficient enough unless the population is highly selected which decreases the external validity.
line 403 :
Thus, it is not impossible in the geriatric field as demonstrated by Gilbert et al study[30].”
Author Response
All comments have been addressed.
Just some minor comments on English language that can be corrected by the authors.
Minor comments:
Line 383 :
Monodomain interventions do not seem to be efficient enough unless the population is highly selected which decreases the external validity.
line 403 :
Thus, it is not impossible in the geriatric field as demonstrated by Gilbert et al study[30].”
We thank the reviewer for the relevant comments on the manuscript.
We modified and updated the manuscript according to the reviewers’ comments and made some corrections on English language.
We sent the manuscript to English language editing through MDPI website for better English proof-reading and editing.
We hope the modifications performed in the manuscript will meet your expectations.
Reviewer 2 Report
Paper much improved - needs minor english language proof-reading and editing
Author Response
Paper much improved - needs minor english language proof-reading and editing
--We thank the reviewer for their relevant comments on the manuscript.
We made some corrections on English language in the manuscript.
We sent the manuscript to English language editing through MDPI website for better English proof-reading and editing.
We hope the modifications performed in the manuscript will meet your expectations.